# Quantifying the Impact of Alternative Bus Stop Platforms on Vehicle Emissions and Individual Pollution Exposure at Bus Stops

**DOI:** 10.3390/ijerph19116552

**Published:** 2022-05-27

**Authors:** Qian Yu, Lili Lu, Tiezhu Li, Ran Tu

**Affiliations:** 1College of Transportation Engineering, Chang’an University, Xi’an 710064, China; yuqian@chd.edu.cn; 2Key Laboratory of Transport Industry of Management, Control and Cycle Repair Technology for Traffic Network Facilities in Ecological Security Barrier Area, Xi’an 710064, China; 3Faculty of Maritime and Transportation, Ningbo University, Ningbo 315211, China; 4Jiangsu Province Collaborative Innovation Center for Modern Urban Traffic Technologies, Nanjing 211189, China; litiezhu@seu.edu.cn (T.L.); turancoolgal@seu.edu.cn (R.T.); 5School of Transportation, Southeast University, Nanjing 211189, China

**Keywords:** emissions, bus stops, portable emission measurement system, traffic simulation, numerical simulation

## Abstract

Due to stop-and-go events, bus stops are often treated as “hot spots” for air pollution. The design of bus stops should be optimized to reduce emissions and exposure for transit commuters. The objective of this study was to analyze the impact of bus stop platform types on vehicle emissions and individual pollution exposure. Second-by-second emissions data were first collected from one bus using a portable emission measurement system (PEMS). Microscopic traffic simulation was then used to estimate overall traffic emissions under six scenarios with different bus stop settings. Numerical simulation of pollutant dispersion was also conducted to calculate individual pollution exposure at bus stops. The results of PEMS tests showed no significant differences between bus emissions generated near two different types of stops. However, the effect of platform types on overall traffic emissions was revealed using traffic simulation. The results demonstrated that bus bays reduced the emissions of other heavy-duty vehicles. However, bus bays were not always effective during rush hours. The study also highlighted the importance of the location of bus stops, the number of bus lines, and the length of the platform, in addition to dynamic characteristics of traffic flows in the design of bus stop platforms. Bus stop platforms also affected individuals’ exposure due to the changes in the pollutant flow field. The passenger’s exposure at one bus stop was influenced by both the platform type and standing location. Results suggested that in a condition with a wind direction perpendicular to the bus stop shelter, the total exposure level to CO was lower at the bus bay stop if a passenger stood at the upstream of the station platform. However, the exposure was less at the downstream of the curbside bus stop.

## 1. Introduction

Fuel consumption (FC) and emissions on a segment of road increase considerably when stop-and-go traffic conditions occur [1,2], and because of that, overall traffic emissions near bus stops are always higher than those on other road segments. A previous study showed that emissions generated at those locations accounted for approximately 20% of the total volume of emissions in a trip of a bus [3]. While waiting at bus stops, transit commuters may be exposed to a large amount of traffic-related air pollutants in relatively short periods of time due to their proximity to the roadway [4,5], posing health risks to commuters [6]. Therefore, it is important to reduce emissions and personal exposure at bus stops.

Several studies were conducted to accurately quantify bus emissions in order to improve the planning and design of the transit system. The research data were obtained mainly using a portable emission measurement system (PEMS) and/or traffic simulation. Wang et al. developed a composite line source emission model to quantify the spatial variability of vehicle emissions in a traffic-interrupted microenvironment [6]. They found emissions were highest when buses were accelerating. Li et al. investigated the impact of bus stop location on emissions [7]. They found that the application of transit signal priority (TSP) for buses approaching far-side stops was another efficient solution to reduce emissions. Alam et al. investigated possible transit service improvements for reducing bus greenhouse gas (GHG) emissions along a busy transit corridor [8]. The results showed that express bus service and smart fare payment cards could reduce total bus emissions. Chen et al. analyzed the effectiveness of a proposed stop-skipping strategy, and showed it could outperform other operating strategies in terms of bus emission reduction [9]. Wang et al. compared the change in bus speed, acceleration, and emissions between bus stops, intersections, and road sections by applying statistical methods. The result suggested that bus stops influence traffic operations and emissions [10]. Waraich et al. simulated transit bus ridership and GHG emissions in a network of 200 bus lines [11]. The study found that bus lines with very low ridership contributed to high per capita emissions, and bus routes with high ridership and low per capita emissions could benefit from increased bus frequency. Yu et al. introduced a method to customize the bus system and provided planning suggestions for bus lines and stops based on massive demand data. Moreover, the emission reduction potential of the designed bus lines was also analyzed [12]. Rosero et al. investigated the effects of passenger load, road grade, and congestion level on bus emissions in Madrid, Spain [13]. The research objectives and methods of previous studies are described in Table 1.

The impact of different designs at bus stops on traffic operation and safety has also been explored [16,17,18], which showed that alternative bus stop platforms could be beneficial to mitigating traffic congestion and achieving safer operations. However, emissions and individual exposure near bus stops should be further considered in the design of the bus stop platform.

The purpose of this study was to analyze emissions generated near different bus stops and investigate the impact of bus stop platforms on emissions and individual exposure. First, PEMS tests were conducted to collect real-world bus emissions and FC data, and to explore the operation and emission characteristics of a diesel bus near different bus stops. Second, to analyze the impact on overall traffic emissions on roads, a corridor in an urban area was selected for comparative analysis by micro-traffic simulation. Different scenarios with different bus stop configurations were designed, and emissions were estimated using traffic simulation data and emission models. The impact of bus stop platforms on overall traffic emissions was also quantified in this paper. Then, the influence of platform type on individual exposure was analyzed based on numerical simulation.

The paper is organized as follows. Section 2 presents the methods in this paper, including the scheme of PEMS tests, development of the driving cycle, traffic simulation and emission models, and pollution dispersion simulation. Section 3 discusses the results based on the PEMS test. Section 4 discusses the micro-traffic simulation results in different scenarios. Section 5 analyzes the numerical simulation results of individual exposure at bus stops. Finally, Section 6 concludes this paper and gives future research directions.

## 2. Methods

To simplify this research, bus stops with dividers, which separate motor and non-motor vehicles, were chosen as research objects. Figure 1 shows the two types of bus stops investigated in this study. Two methods were applied to analyze CO_2_, CO, NO_X_, and HC emissions at bus stops. First, on-road PEMS tests were conducted to collect the emissions data of buses during normal running in Nanjing, China. The data were used to analyze the influence of bus stop platforms on individual bus emissions. Second, microscopic traffic simulation with an instantaneous speed-based emission model was applied to calculate the total traffic emission in different scenarios. The influence of alternative bus stop platforms on the operation and emissions of other on-road vehicles were examined. Finally, the influence of platform type on individual exposure was analyzed based on numerical simulation.

### 2.1. PEMS Tests

SEMTECH-DS, a PEMS equipment manufactured by Sensors Inc., Saline, MI, USA, was used to obtain real-world emissions near both types of bus stops, as shown in Figure 2. Vehicle speeds, carbon dioxide (CO_2_), carbon monoxide (CO), nitrogen oxides (NO_x_), and hydrocarbon (HC) exhaust concentration, and corresponding emission rate data were logged for each second. The fuel consumption (FC) rate was calculated using the carbon balance method. The measured buses were equipped with an in-line 6-cylinder, 8.27 L direct injection diesel engine. The rated horsepower of the engine was 184 kW at 2200 r/min. To eliminate the impact of different driving behaviors of different drivers, field data collection was conducted with one normal driver in Nanjing, China. Passengers boarded and alighted at bus stops.

The selected bus route (bus line #100 in Nanjing) was 13.4 km long. Before using PEMS, a field survey was carried out at bus stops to record the location, the type of bus stop platform, and the length of the bus stop. In total, 33,197 valid second-by-second data points were collected by PEMS. Out of those, the data of 41 bus stops that met the following research conditions were selected. Twenty-two were curbside bus stops, and nineteen were bus bays [3].

Data obtained from the same stop at different times were treated as independent samples due to different traffic conditions. The number of passengers, queuing near bus stops, arrival and departure time, and delay at intersections were recorded manually during tests to facilitate analysis. Finally, two independent groups of real-world emissions and FC data for bus bay and curbside stops were obtained. The effects of bus stop platforms on emissions and FC near bus stops were analyzed by non-parametric tests, including the Mann–Whitney (MW) test and Kolmogorov–Smirnov (KS) test. The null hypothesis that corresponded to the situation was that there would be no differences between the medians and distributions of the two groups of data. If the *p*-value was smaller than the level of significance, the null hypothesis would be rejected.

To analyze the influence of bus stop types on bus operation and emissions, driving cycles near the two types of stops were developed, respectively. The micro-trip method was employed to develop driving cycles [19,20]. The method followed five steps: extracting bus stop micro-trips from the collected data, selecting 12 driving parameters to describe the driving characteristics of micro-trips, generating four principal components by principal component analysis, clustering micro-trips into two groups (during peak-hour or off-peak hour), and developing a driving cycle. The twelve driving parameters are listed in Table 2. The optimal driving cycle was determined by minimizing the average errors of the 12 driving parameters between the constructed driving cycle and the collected real-world driving data.

### 2.2. Microscopic Traffic Simulation and Emission Models

#### 2.2.1. Microscopic Traffic Simulation

Since it was difficult to simultaneously obtain emission data from other vehicles near the bus stops during PEMS tests, microscopic traffic simulation was used to collect traffic data for analyzing the emissions of other fleets. Transit data were imported through two steps using VISSIM [21]:

Step 1: Define bus stops and select the type of bus stop platform;

Step 2: Import bus routes and schedules.

The microsimulation was conducted for Longpan Road in Nanjing, China, with five cross-streets. The network consisted of 221 links, 5 signal controllers, and 14 bus stops (seven stops at the northbound direction and southbound direction, respectively), as shown in Figure 3. In order to consider other factors, such as the location of bus stops, the number of bus lines, the length of the platform, and dynamic characteristics of traffic flows in the design of bus stop platforms, 14 stops were selected to analyze emissions.

The length of the modelled road was about four kilometers, with three or four lanes in each direction. Traffic volumes and the proportion of turning movements (left and right turns) at each intersection were counted using the real-time video data. Three types of vehicles were simulated, namely, light-duty gasoline vehicles (LDGVs), diesel buses, and other heavy-duty diesel vehicles (HDVs) such as coaches. During peak hours, the volume of southernmost road sections was 1675 veh/h in the northbound direction, and the ratio of LDGVs, buses, and HDVs was about 1578:77:20. Five traffic lights were controlled with the traditional fixed-time method, and real-world signal timings were collected at each intersection. Eighteen bus routes were distributed along this road, and the bus schedules for every line during the peak morning period, in addition to the total boarding and alighting time at each stop, were obtained through a field survey. The simulation lasted for 3600 s. Base case data were collected during peak morning hours (7:30 AM–8:30 AM). The type of bus stop platform, the length of the bus stop, the number of served bus lines, and the location of the bus stop are shown in Table 3.

Six different scenarios, including the base case, were simulated in consideration of bus stop location, and the total number of bus lines served, to understand the influence of bus stop platforms on vehicle emissions. Emissions were compared to identify the impact of the alternative types of bus stop platforms. The different scenarios are described as follows:Scenario 1: all stops were bus bay stops;Scenario 2: all stops were curbside stops;Scenario 3: two mid-block bus stops in the central business district (CBD) (No. 4 and No. 11) were changed to curbside stops;Scenario 4: as a base case, all 14 bus stops were set according to situations in Table 1;Scenario 5: two far-side bus stops (No. 7 and No. 8) were changed to bus bay stops;Scenario 6: two bus stops (No. 2 and No. 13) that were serving most lines, were changed to bus bay stops.

#### 2.2.2. Emission Models

The comprehensive modal emission model (CMEM), a physical power–demand model, was adopted to estimate emissions. CMEM was developed based on engine load and the physical–chemical theories of emissions, which can reach higher accuracy than traditional methods based on statistics and parameter estimation. The core model of CMEM needs two input files to estimate emissions and FC on a second-by-second basis: one defines the type of vehicle and the parameters for the model calculation, while another provides second-by-second vehicle activities [22,23]. In this study, CMEM was used in combination with VISSIM to estimate emissions of the fleets under different scenarios. The fleet composition was determined at every link based on traffic surveys, and parameters for each type of vehicle in CMEM are presented in Table 4.

To validate the estimation results from CMEM, emissions of CO_2_, CO, NO_X_, and HC from buses were estimated by CMEM and compared with the PEMS record. Absolute relative errors of total bus emissions were 9.3%, 7.4%, 8.1%, and 5.8%, for CO_2_, CO, NOx, and HC, respectively. CMEM can be used to obtain relatively accurate emission estimation results.

### 2.3. Pollution Dispersion Simulation and Inhalation Estimation at Bus Stops

In order to compare the effects of different platform types on individual exposure, the diffusion of one 10 m long diesel bus emission was simulated, based on the k-ε turbulence model in this paper. An analysis as to whether different bus stop settings affected the diffusion feature of pollutants and led to changes in individual exposure and inhalation, was undertaken. A three-dimensional model, where the X-axis originates from south to north, and the Y-axis originates from east to west, as shown in Figure 4, was used to simulate the microenvironment at bus stops. The forward direction of the bus at a curbside bus stop was parallel to the positive direction of the Y-axis. The height of waiting passengers was set to 1.65 m. The boundary condition of the physical model is as follows in Table 5.

There were no chemical reaction processes during the simulation. The renormalization group (RNG) k-ε model was utilized. The constant values in the model were set according to the literature [15]. The numerical calculation was performed in Ansys Fluent 18.0. The outputs of the numeric simulation were CO concentration values (volume fraction per second) on the breathing surface of waiting passengers. The comparison and subsequent inhalation calculation were carried out for the duration of one bus travelling along the bus stop platform. The total exposure was calculated according to the following equation [5,24]:(1)$$E_{total}=\int_{t_{1}}^{t_{2}}\left(C_{i}\left(t\right)\times IR_{i}\right)dt$$
where, *E_total_* is the total exposure for the duration of one bus travelling along the bus stop (g); *t*_1_ and *t*_2_ are the starting and ending times of exposure to bus emissions, respectively; *C_i_*(*t*) is the individual exposure due to bus emissions at time *t* (g m^−3^); *IR_i_* is the individual inhalation rate (=1.73 × 10^−4^ m^3^ s^−1^), which was derived from the United States EPA’s exposure factors handbook [24].

## 3. The Impact of the Bus Stop Platform on Bus Emissions

Data collected within stop influencing zones were divided into three driving modes, namely, deceleration, idling, and acceleration. Based on previous research, it was known that emissions during idling mostly depend on the number of passengers boarding or alighting the bus [3]. Therefore, non-parametric tests were mainly applied to analyze the impact of acceleration and deceleration on emissions and FC near bus stops.

Table 6 shows the non-parametric test results based on real-world data. The *p*-values of both tests were greater than 0.05, indicating no significant difference between both types of bus stops for the four pollutants: the results of the MW test showed the difference between medians of both types were negligible, while the KS test showed emissions and FC generated near bus bays and curbside bus stops conformed to the same distribution. From these results, it was concluded that bus stop platforms did not significantly affect bus emissions and FC. In addition, the platform type also had no significant influence on the emission generated during acceleration and deceleration.

Based on the method mentioned at Section 2.3, two driving cycles were then developed for the two types of bus stops based on real-world driving data. Principal component analysis was performed for the 12 driving parameters. Four principal components were extracted, and the cumulative variance proportions were 88.14% and 86.56%, respectively. The micro-trips with minimum error for the 12 parameters were picked out for peak hour and off-peak hour. The developed driving cycles of bus bays and curbside bus stops are shown in Figure 5.

As shown in Figure 5, for both types of bus stops, the bus driver always maintained a low speed when leaving the station and the bus always encountered a queue and a second stop during peak hours. As the bus arrived, idled, and departed from a bus stop, the driving cycles were essentially the same for the two types of bus stops. Therefore, the bus stop platform type did not have a significant effect on bus operation and bus emissions in this case study.

## 4. The Impact of the Bus Stop Platform on Overall Traffic Emissions

Although emissions from the two types of bus stops did not display a significant difference in Section 4, bus bay stops may not have had as many effects on the surrounding road traffic as curbside bus stops, since bus bay stops interfered with other passing vehicles only when buses maneuvered to pull into and out of the stop. Therefore, differences may exist between bus stops in terms of the interaction of buses with other vehicles, which leads to different amounts of emissions from other on-road fleets. In this section, different scenarios with varied bus stop platform designs are examined, and the result is compared with the effect of the platform design.

Emissions data under different scenarios were obtained by combining VISSIM and CMEM. The calculation of emissions in different scenarios was based on second-by-second speed data from every vehicle in traffic simulation. The total number of vehicles and distance travelled within one hour in different scenarios is shown in Table 7. Results for total emissions (g) and distance-based emission factors (g/km) under the different scenarios were compared. A summary of the total emissions and percentage differences (compared with the base case scenario 4) for CO_2_, CO, NO_X_, and HC is provided in Table 8. Changes were small, with percentage differences below 3%.

Figure 6 illustrates the total emissions of LDGVs, diesel buses, and other HDVs. The fleet of LDGVs showed a major contribution to total emissions of CO_2_, CO, and HC during peak hours, while diesel buses contributed most to NO_x_ emissions. Figure 7 shows the percentage difference between the distance-based emission factors of various scenarios and the base case. In addition, the average emissions (g/vehicle) of different fleets were calculated. Because the traffic volume of the three fleets did not change significantly among different scenarios, the trends of the average emissions were consistent with total emissions. This also meant that during rush hours, the capacity of the road network did not increase by changing the type of bus stop platform.

In scenario 1, all bus stops were set as bus bay stops. Compared with the base case, emissions from heavy-duty diesel vehicles reduced by about 19%. Bus bays reduced the environmental impact on heavy-duty diesel vehicles in this case; however, they did not reduce emissions of the whole fleet, with a 2.65% increase for CO_2_ and 0.15% increase for CO. In fact, total emissions of CO_2_ from the fleet of LDGVs increased by 4.2% and emission factors (g/km) for the four gaseous emissions all increased the same. However, overall traffic emissions were not significantly different from scenario 4 (base case). Bus bay stops did not have a big impact during rush hours on emissions. Setting all bus stops to bus bays was insufficient to increase the capacity of the road network. On the contrary, setting bus bays with the same length may reduce berths, and buses must queue when entering the station during peak hours, leading to higher emissions and a negative impact on traffic operations.

In scenario 2, all bus stops were curbside. Unlike bus bay stops, curbside bus stops in scenario 2 increased emissions from HDVs, which were the highest among all scenarios. However, car emissions did not increase significantly, and bus emissions showed no significant reduction.

In scenario 3, two mid-block bus stops (No. 4 and No. 11), located in the CBD, were changed to curbside stops. Bus bay stops are usually selected for such locations. After changing bus stops No. 4 and No. 11 to curbside stops, the emission factor of bus fleets decreased slightly, and emissions from HDVs dropped slightly. Although these two stations were located in the CBD, where traffic demand is high, the type of platform did not have a great impact on the traffic near the bus stops. These two bus stops were far away from intersections, and they served only four bus lines so they were less likely to suffer traffic jams caused by buses pulling into and out of the stop. A bus bay was not necessarily needed for these locations.

In scenario 5, two far-side bus stops (No. 7 and No. 8) were changed to bus bay stops. The overall change in total emissions was small, within 3%. Changing to bus bays did not reduce emissions as expected. Emissions of fleets around far- and near-side bus stops were also influenced by signal control and geometric design due to the complexity of traffic around intersections. The design of the bus stop needed to be combined with signal timing optimization at intersections to achieve smoother traffic and fewer emissions [7,25].

In scenario 6, two bus stops (No. 2 and No. 13) were changed to bus bay stops. These two stations served up to 10 routes, the largest number of bus routes. The emission factor of HDVs decreased after changing the bus stop, effectively reducing the impact on large vehicles. There was a slight increase in the emission factor for the bus fleet. It is worth noting that except for CO, the emission factors of car fleets increased due to setting bus bays. The main reason was that the short length of the bus stop platform (22 m) resulted in a maximum of one berth when setting a bus bay. It may cause a queue of buses waiting to enter the platform during peak hours.

## 5. The Impact of the Bus Stop Platform on Individual Exposure

Related experimental studies have shown that factors influencing pollutant exposure at bus stops include time of day, locations where passengers are waiting, land use near bus shelters, and smoking at bus shelters [26]. In this study, other relevant influencing factors were controlled, only the type of platform was changed, and the pollutant diffusion was simulated when a diesel bus arrived, idled, and departed from the bus stop. The exposure concentration of passengers at different locations were compared. The results showed that a change in platform type will cause fluctuating pollution levels upstream and downstream of the bus shelter. The emission emitted by the bus increased the concentration of pollutants within the bus stop area and changed the distribution of pollutants along the platform. Even if the emission source intensity and other diffusion conditions, in addition to passengers’ waiting location, were the same, the bus bay led to changes of the movement angles when a bus entered and left the bus stop. It affected the flow distribution between the bus and the shelter, and then changed the trajectory of pollutant diffusion. As shown in Figure 8, when the bus left the platform, the flow field changed due to the bus bay, resulting in the difference of exposure concentration at the bus stop.

Table 9 lists the total individual CO exposure at different locations of different types of platforms during the service time of one bus. Based on the results of the numerical simulation, it was found that the platform type affected the pollutant diffusion in the simulated wind environment. The total exposure varied at different locations of the bus stop. During the service time of one diesel bus, the exposure upstream of the station (position 5) increased by 5.29–11.05 times compared with downstream (position 1). At the same location, the total exposure of different platform types was also different. The total exposure at the bus bay stop was less than that of a curbside bus stop if one passenger stood upstream or at the middle of the station platform, while there was less pollutant exposure downstream of the curbside bus stop compared with a bus bay stop.

## 6. Conclusions and Future Work

This paper first investigated bus emissions generated around stops based on PEMS data. VISSIM and CMEM were also used to simulate and estimate traffic emissions from three types of vehicles (LDGV, bus, and HDV) under different scenarios in a major roadway in Nanjing, China. A numerical simulation was then conducted to analyze the exposure at two different bus stop types. The results showed that:Bus stop platforms barely affected emissions and FC of the bus itself. Optimizing bus stop platform design did not improve bus emissions, FC, or operations in this case;Bus bays may not work well during rush hours. With insufficient berths, bus bays can also lead to queues that obstruct cars behind buses;A comprehensive analysis was required for bus stop location, the number of bus lines, the length of the platform, and the dynamic characteristics of traffic flow. Thorough traffic surveys and analyses were vital before deciding on which type of bus stop was more suitable;Even if the intensity of pollution sources and diffusion conditions are all the same, the level of exposure at different types of platforms showed a large difference.

In this study, PEMS tests were only conducted on limited bus routes because of the heavy workload required for basic data collection of bus stops. For future work, this study could be extended to other types of buses (such as the long-articulated bus), other drivers, and under other traffic conditions. More PEMS tests should be conducted to verify and generalize the trends discovered by this study.

At present, the electrification of vehicles is an important measure to solve the problem of vehicle emissions. This paper only studies the vehicles with traditional fossil fuels. In the future, the emission, power consumption, and equivalent carbon emission can be further analyzed, which are caused by the surrounding traffic within the influence zone of bus stops.

The traffic simulation was only conducted on one major roadway with a specific geometric design and traffic conditions. To generalize findings, this simulation could be extended to different locations, roadways, lanes, bus stop lengths, fleet compositions, and traffic flows. Additionally, some parameters of the car-following and lane-change models were set using recommended values from other literature. Calibration of the car-following model and lane-change model in VISSIM should be further conducted if real-world data are available.

In addition, the influence of wind speed and direction and the background concentration of CO in the air were not taken into account in the study of pollutant diffusion at the bus stops. Further quantitative analysis can be carried out in combination with actual microenvironment monitoring and more influencing factors.

## Figures and Tables

**Figure 1 ijerph-19-06552-f001:**
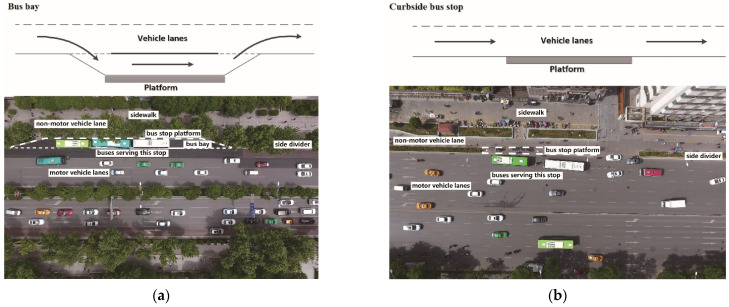
The two types of bus stop in this study: (**a**) bus bay stop (BBS), (**b**) curbside bus stop (CBS).

**Figure 2 ijerph-19-06552-f002:**
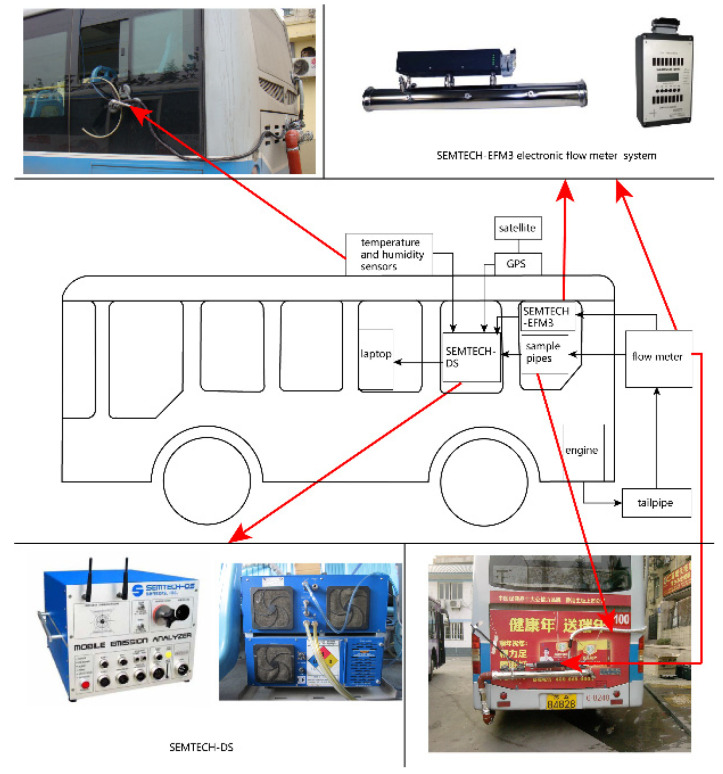
On-road PEMS test equipment.

**Figure 3 ijerph-19-06552-f003:**
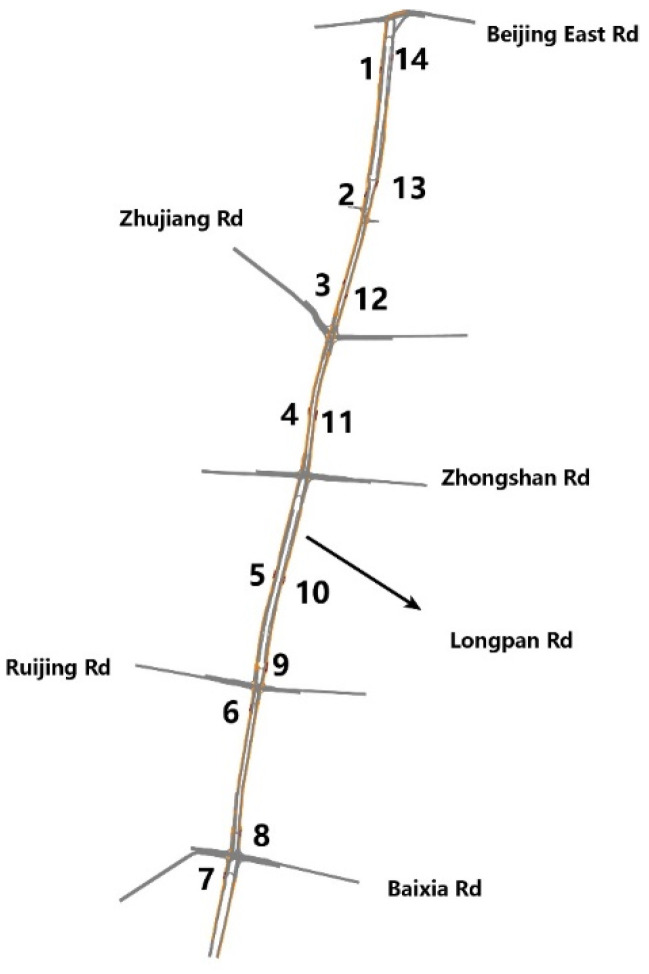
Location of the 14 stops along Longpan Road.

**Figure 4 ijerph-19-06552-f004:**
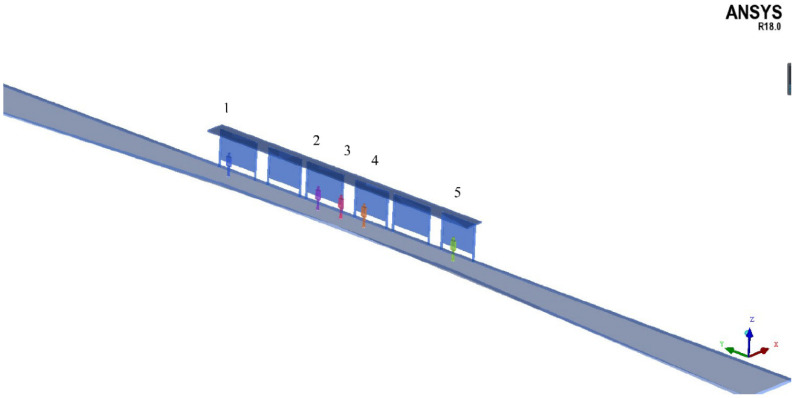
Simulation of the microenvironment and assumed passenger standing points at one bus stop.

**Figure 5 ijerph-19-06552-f005:**
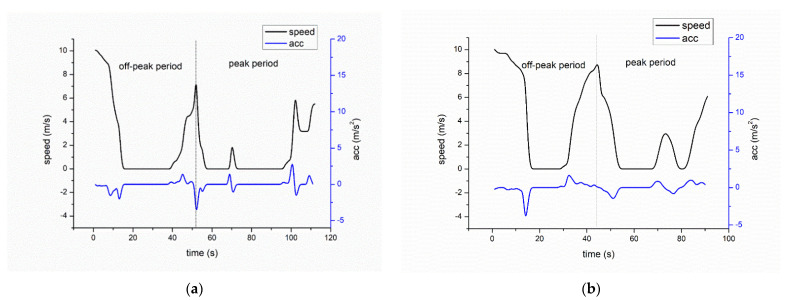
Driving cycle for buses near the two types of bus stops: (**a**) developed driving cycle near bus bay, (**b**) developed driving cycle near curbside bus stop.

**Figure 6 ijerph-19-06552-f006:**
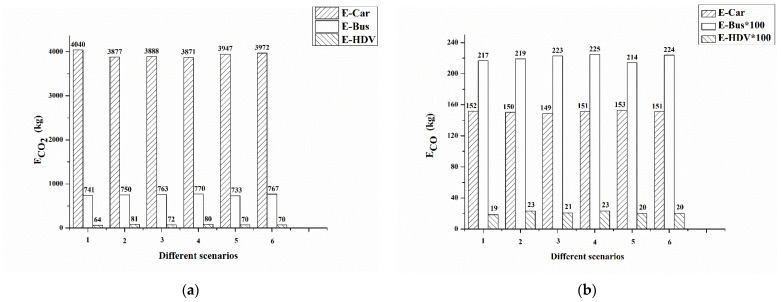
Total CO_2_, CO, NO_X_, and HC emissions for light-duty gasoline cars, diesel buses, and other HDVs under different scenarios: (**a**) CO_2_, (**b**) CO, (**c**) NO_X_, and (**d**) HC. Note: E-Car, E-bus, and E-HDV denote the total emissions of light-duty gasoline cars, diesel buses, and other HDVs, respectively.

**Figure 7 ijerph-19-06552-f007:**
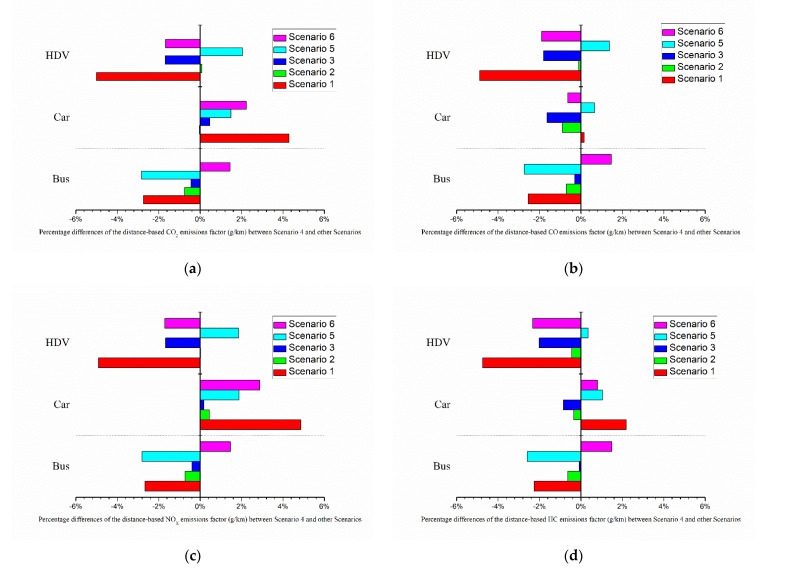
Percentage differences of emission factors in different scenarios compared to base case: (**a**) CO_2_, (**b**) CO, (**c**) NO_X_, and (**d**) HC.

**Figure 8 ijerph-19-06552-f008:**
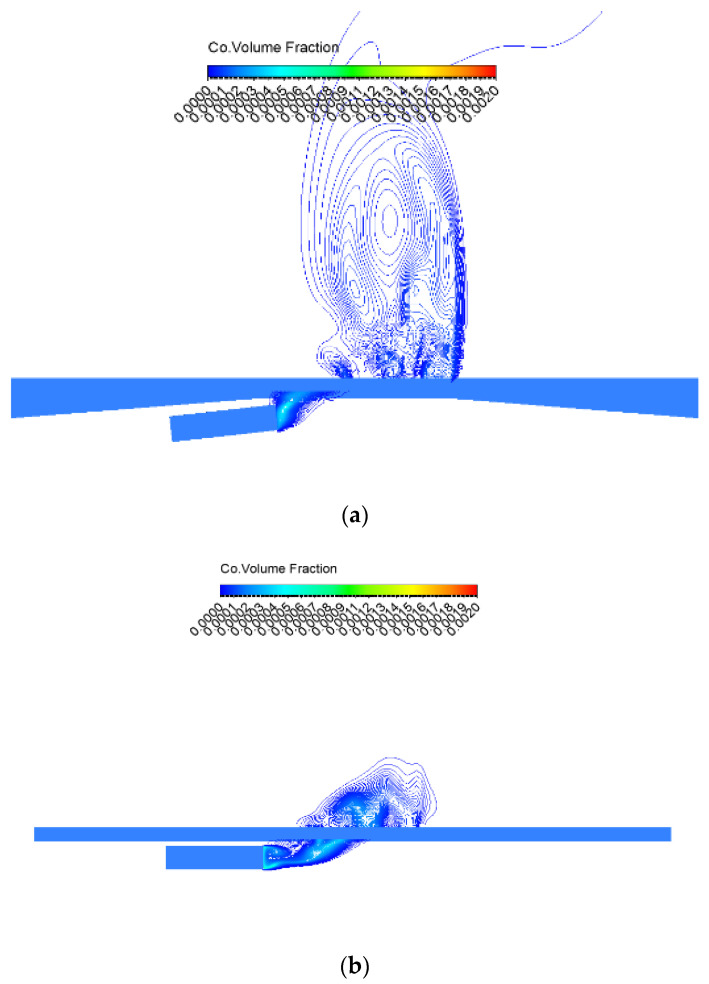
CO diffusion simulation when the bus leaves the station: (**a**) diffusion simulation of CO emission at bus bay stop, (**b**) diffusion simulation of CO emission at curbside bus stop.

**Table 1 ijerph-19-06552-t001:** Summary of previous studies on bus emissions.

Author (Year)	Study Focus	Method
Wang et al., (2010) [6]	The spatial variability of vehicle emissions in a traffic-interrupted microenvironment.	Composite line source emission model.
Li et al., (2012) [7]	Bus stop location and emissions.	Traffic simulation and VSP bin model.
Alam et al., (2014) [8]	Transit service improvements for reducing bus greenhouse gas (GHG) emissions along a busy transit corridor.	Traffic simulation.
Chen et al., (2017) [9]	Stop-skipping strategy and bus emission reduction.	Traffic simulation and VSP bin model.
Wang, (2017) [14]	Pollutant diffusion and dispersion at an urban bus stop.	Microscopic diffusion model, directly measuring the concentration.
Dou et al., (2018) [15]	The exposure at the traffic microenvironment.	Computational fluid dynamics numerical simulation.
Wang et al., (2018) [10]	Bus speed, acceleration, and emissions between bus stops, intersections, and road sections.	PEMS tests.
Waraich et al., (2019) [11]	Transit bus ridership and GHG emissions.	Traffic simulation.
Yu et al., (2020) [12]	Customization of bus system and the potential to reduce emission.	Traffic simulation.
Rosero et al., (2021) [13]	Passenger load, road grade, and congestion level and emissions.	PEMS tests.

**Table 2 ijerph-19-06552-t002:** Driving parameters of micro-trips at bus stops.

Parameter	Abbreviation	Definition	Unit
1. Maximum speed	V_max_	Maximum speed of entire trip	km h^−1^
2. Average speed	V_m_	Average speed of entire trip	km h^−1^
3. Average running speed	V_mr_	Average speed of entire trip excluding idle time	km h^−1^
4. Maximum acceleration	A_max_	Maximum acceleration of entire trip	m s^−2^
5. Maximum deceleration	A_min_	Maximum deceleration of entire trip	m s^−2^
6. Average acceleration	A	Average acceleration of entire trip	m s^−2^
7. Average deceleration	D	Average deceleration of entire trip	m s^−2^
8. speed variance	*V_s_*	Speed variance of entire trip	km^2^ h^−2^
9. Running time	T	Running time of entire trip	s
10. Acceleration proportion	P_a_	Percentage of time spent in acceleration mode	%
11. Deceleration proportion	P_d_	Percentage of time spent in deceleration mode	%
12. Idling proportion	P_i_	Percentage of time spent in idling mode	%

**Table 3 ijerph-19-06552-t003:** Characteristics of bus stop along the selected corridor.

Stop Number	Platform Type	Platform Length (m)	Served Bus Lines Count	Bus Stop Location
1	CBS ^a^	25	8	far-side
2	CBS	22	10	mid-block
3	CBS	15	6	near-side
4	BBS ^b^	28.5	4	mid-block
5	BBS	30	5	mid-block
6	CBS	27	5	far-side
7	CBS	25	8	far-side
8	CBS	27	6	far-side
9	BBS	40	5	far-side
10	BBS	30	5	mid-block
11	BBS	37	4	mid-block
12	CBS	15	6	far-side
13	CBS	25	10	mid-block
14	CBS	25	8	near-side

Note: ^a^ curbside bus stop; ^b^ bus bay stop.

**Table 4 ijerph-19-06552-t004:** Inputs used for emissions estimation in CMEM.

Parameters	Diesel Buses	Other HDVs	LDGVs
Vehicle mass (kg)	9450	10,450	1250
Engine displacement (L)	6.7	8.3	2
Engine torque at maximum power (N.m)	998	1050	163
Engine speed at maximum power (rpm)	2500	2500	6000
Maximum power (kW)	188	184	110
Engine speed at maximum torque (rpm)	1700	1700	4000
Number of gears	5	5	4
Number of vehicles of this type (base case)	1123	97	14,194

**Table 5 ijerph-19-06552-t005:** Boundary condition of the physical model.

Boundary	Boundary Conditions
Inlet	Velocity-inlet, v = 4.5 m/s, wind direction is perpendicular to bus stop shelter and parallel to the positive X-axis, temperature = 300 k, CO mass fraction = 0 ^a^
Outlet	Pressure-outlet, static pressure = 0 pa
Pollutant Inlet	Velocity-inlet, v = 4.8 m/s, temperature = 300 k, CO volume fraction = 0.017% ^b^
Top and sidewall	Symmetry
Ground	Stationary wall, roughness = 0

Note: ^a^ the simulation ignores the concentration of CO in the air; ^b^ CO fraction of bus exhaust was calculated according to the mean value of CO concertation during PEMS tests.

**Table 6 ijerph-19-06552-t006:** Non-parametric test results for bus emissions and FC at the two types of bus stops.

Emission or FC (g)	Bus Stop Type	Total Emissions Generated Near the Bus Stop						Emissions Generated during Acceleration		Emissions Generated during Deceleration	
		Mean	Std.	Min.	Max.	*p*-Value		*p*-Value		*p*-Value	
						(α = 0.05)		(α = 0.05)		(α = 0.05)	
						MW-Test	KS-Test	MW-Test	KS-Test	MW-Test	KS-Test
CO_2_	Curbside	88.16	35.34	36.12	138.81	0.95	0.85	0.35	0.12	0.52	0.66
	Bus bay	87.31	34.65	34.36	157.81						
CO	Curbside	0.41	0.19	0.10	0.70	0.53	0.56	0.06	0.11	0.23	0.11
	Bus bay	0.37	0.16	0.11	0.63						
NO_X_	Curbside	1.62	0.57	0.76	2.42	0.75	0.95	0.35	0.13	0.56	0.64
	Bus bay	1.68	0.62	0.78	2.88						
HC	Curbside	0.05	0.01	0.03	0.08	0.99	0.54	0.24	0.11	0.45	0.77
	Bus bay	0.05	0.02	0.03	0.09						
Fuel consumption	Curbside	28.04	11.24	11.46	44.19	0.95	0.85	0.15	0.22	0.52	0.66
	Bus bay	27.75	11.01	10.91	50.13						

**Table 7 ijerph-19-06552-t007:** Total number of vehicles and distances travelled within one hour.

Scenario	LDGVs		Buses		HDVs	
	Counts	Distance (km)	Counts	Distance (km)	Counts	Distance (km)
1	14,194	18,103	1123	1580	97	117
2	14,236	18,125	1133	1567	97	139
3	14,251	18,086	1133	1588	97	126
4	14,247	18,090	1133	1596	97	137
5	14,235	18,176	1132	1564	97	118
6	14,245	18,157	1133	1568	97	122

**Table 8 ijerph-19-06552-t008:** Summary of total emissions in the simulated network and percentage differences to the base case under different scenarios.

Emissions	Scenario 1	Scenario 2	Scenario 3	Scenario 4	Scenario 5	Scenario 6
E_CO2_ (kg)	4846.15	4708.25	4722.82	4721.00	4749.79	4809.11
△E_CO2_ ^a^	2.65%	−0.27%	0.04%	0.00%	0.61%	1.87%
E_CO_ (kg)	154.01	152.64	151.21	153.78	155.37	153.35
△E_CO_	0.15%	−0.74%	−1.67%	0.00%	1.03%	−0.28%
E_NOX_ (kg)	15.79	15.78	15.80	15.96	15.58	16.02
△E_NOX_	−1.04%	−1.07%	−0.99%	0.00%	−2.34%	0.43%
E_HC_ (kg)	3.47	3.45	3.44	3.48	3.45	3.49
△E_HC_	−0.15%	−0.84%	−1.06%	0.00%	−0.89%	0.28%

Note: ^a^ percentage differences compared to the base case (positive sign indicates an increase in emissions).

**Table 9 ijerph-19-06552-t009:** Total exposure at different types of bus stops.

Passenger Location ^a^	5 (The Upstream of the Bus Stop Platform)	3 (The Middle of the Bus Stop Platform)	1 (The Downstream of the Bus Stop Platform)
*E_total_* per bus trip at BBS ^b^ (g)	4.39 × 10^−5^	3.59 × 10^−6^	6.98 × 10^−6^
*E_total_* per bus trip at CBS ^c^ (g)	6.53 × 10^−5^	9.57 × 10^−6^	5.42 × 10^−6^
△E _total_ ^d^	49%	166%	−22%

^a^ The position label is shown in Figure 3; ^b^ bus bay stop; ^c^ curbside bus stop; ^d^ positive numbers represent less exposure at bay bus stops.

## Data Availability

The raw/processed data required to reproduce these findings cannot be shared at this time as the data also forms part of an ongoing study.

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
