# Peer review of "Quantifying the Impact of Alternative Bus Stop Platforms on Vehicle Emissions and Individual Pollution Exposure at Bus Stops"

_ijerph, 2022, doi:10.3390/ijerph19116552_

Round 1

Reviewer 1 Report

Quantifying the Impact of Alternative Bus-Stop Platforms on Vehicle Emissions and Individual Pollution
Exposure at Bus Stops

1. Improve the image quality of the figures like Figure 2.
2. In ‘Introduction’ section it is better to add a table to present the comparison of previous studies.
3. Add some latest updated work in this table.
4. Add some Research Methodology diagram in section 2 ‘Methods’
5. In “Data Availability Statement:” no statement is available.
6. Proof reading will make paper more understandable.

Reviewer 2 Report

Strengths: The article is well prepared, the results are clearly presented, the research methods used are correct. Numerous figures and tables are presented, which enrich the article and allow better understanding of its content and familiarization with the research carried out. Weaknesses: The results of the research confirm the assumptions that could have been made without conducting this research. In conclusion, the article is suitable for publication in a scientific journal.

Reviewer 3 Report

In this study, the effects of Alternative Bus-Stop Platforms on Vehicle Emissions and Individual Pollution Exposure at Bus Stops were investigated. On the whole, this is a relatively complete study, and all methods and analysis results are scientific. Therefore, the authors only need to do some minor revisions according to the following suggestions:

  1. In Figure 1, the authors show two different subjects (BBS and CBS). Are they representative? Why did the authors choose them as the research object?
  2. The reason why 14 stops are selected as research parameters in Figure 2 should also be described in detail in this article.
  3. Generally, the unit of emissions is the concentration used, such as ppm. However, in this study, the author used the unit of weight (kg, g). Why?
  4. English grammar needs double check in this article.
